# Choline in Pregnancy and Lactation: Essential Knowledge for Clinical Practice

**DOI:** 10.3390/nu17091558

**Published:** 2025-04-30

**Authors:** E. J. Derbyshire

**Affiliations:** Nutritional Insight, Epsom KT17 2AA, UK; emma@nutritional-insight.co.uk; Tel.: +44-(0)-7584-375-246

**Keywords:** brain development, choline, clinicians, habitual intakes, health claims, liver function, recommendations

## Abstract

Background/Objectives: In 1998 choline was identified as an essential nutrient by the United States Institute of Medicine. Choline is known primarily for its roles in neurotransmitter production, cell membrane formation, and methyl and lipid metabolism. Since this discovery the relevance of choline to maternal, fetal, and infant health has been studied intensively. This narrative review provides a coherent update of the latest evidence for field clinicians and healthcare professionals. Methods: A PubMed/ScienceDirect search for human clinical evidence restricted to meta-analysis and systematic/review publications from the last 10 years was undertaken. Results: Meta-analysis and review publications highlight the importance of choline in supporting maternal health and fetal development during pregnancy by showing promising roles for choline in relation to neurological development, brain and liver function, reduced neural tube defect risk, and adverse pregnancy outcome risk. However, there are clear present-day gaps between habitual choline intakes and intake recommendations with the majority of pregnant and lactating women not meeting adequate intake recommendations for choline. This gap is anticipated to widen given transitions towards plant-based diets which tend to be lower in choline. Conclusions: Alongside folic acid recommendations, choline supplementation should be considered in dietary recommendations by clinicians during crucial life stages such as pregnancy and lactation when physiological demands for this critical nutrient substantially increase.

## 1. Introduction

The significance of adequate nutrition and micronutrient supplement use in the preconception, pregnancy, and lactation life stages is increasingly being recognized and valued [1,2]. Inadequate nutrient intakes during pregnancy which include choline have been associated with birth defects, altered cognition, motor deficits, visual impairment, heightened risk of abnormal behavior, and conditions such as attention deficit hyperactivity disorder (ADHD), autistic spectrum condition (ASC), anxiety, and depression [3]. The 1910s to 1950s were an era of vitamin discovery and the roles of several single nutrients were first discovered and their roles in deficiency diseases identified [4]. In the 1990s the importance of choline as an essential nutrient was identified from human controlled feeding trials [5]. Formal choline guidance was first compiled in 1998 when it was officially recognized by the Institute of Medicine (IOM) as an essential nutrient [6]. Since then, scientific research has further accrued and choline has been intensively researched for more than 25 years after the publication of this guidance. Despite this there remains a lack of awareness about choline amongst healthcare practitioners and the lay public. Within the scientific literature few articles are written with clinicians in mind, which was the aim of the present article. The objective of the current publication is to provide a coherent update of the latest clinical evidence, choline guidelines, health claims, and practical advice for clinicians. Science has been written succinctly with infographics included to facilitate the dissemination of core information.

## 2. Choline: Functions, Metabolism, and Maternal Needs

### 2.1. Functions

Choline is a critically important water-soluble “essential nutrient”, closely associated with the B vitamins [7,8]. Humans can only produce small amounts of choline in the liver via the hepatic phosphatidylethanolamine *N*-methyltransferase pathway, so it needs to be consumed from food or supplement sources [9].

Choline is needed for brain development, normal liver function, lipid metabolism, and the regulation of homocysteine metabolism [8,10,11]. It is also required for the synthesis of the neurotransmitter acetylcholine, betaine synthesis, and the formation of phosphatidylcholine (PtdCho) [8]. Acetylcholine is the placental storage form for fetal choline supply and important for cognitive development [12]. Sphingomyelin–PtdCho-derived phospholipids (sphingolipids) are required for the myelination of nerve fibers (axons) in the central and peripheral nervous systems [13]. Choline has roles in cell membrane structure and function, methyl-group metabolism, brain development, neuronal differentiation, and lipid transport [14,15,16]. The contribution of choline is essential in the perinatal life stage to assure optimal cognitive development and support neural tube defect (NTD) formation [8].

### 2.2. Metabolism

Choline intake can influence metabolic processes which can affect fetal development, including: (1) epigenetic programming (choline is a methyl donor), (2) neurodevelopment and cognitive function (choline forms acetylcholine and PtCho), (3) placental function (choline can influence functional processes of the placenta), and (4) protection from neural insults [17]. An overview of key metabolic processes is shown in Figure 1.

In the presence of an obesogenic environment metabolic benefits of choline have also been implied [18]. Murine models show that maternal choline supplementation could regulate blood glucose homeostasis [19] and fetal overgrowth [20] and protect the fetal liver from the effects of high-fat diets [21] when offspring are exposed to an obesogenic environment.

### 2.3. Maternal Needs

Pregnancy and lactation are some of the most physiologically demanding life stages that place a unique stress on choline metabolism [22,23]. Maternal choline reserves are depleted during pregnancy and lactation whilst, at the same time, the availability of choline for normal brain development is critical [24]. As shown in Figure 2 the fetus lives in a choline-rich environment. There are substantial demands for choline in late pregnancy, due to enhanced use of choline for PtdCho production [25]. For example, choline levels in amniotic fluid are around tenfold greater than that in the maternal blood [26]. The human placenta also houses around 50 times more choline than maternal blood (≈1000 vs. 20 µmol/L, respectively) [27,28,29] and serum or plasma choline concentrations are six- to sevenfold higher in a fetus or newborn compared with adult levels [26,30,31].

There are also large amounts of choline in breast milk. For example, levels increase by ≈114% from 2–6 days (colostrum phase) to 6–7 days after birth with rises in phosphocholine and glycerophosphocholine accounting for most of this [27,32]. According to the European Food Safety Authority (EFSA) the nutritional needs for choline increase by 20% in pregnancy and 30% in the lactation period [33].

## 3. Approach

A PubMed and ScienceDirect search for human clinical evidence restricted to the last 10 years (meta-analysis and systematic/review publications) was undertaken to identify the latest, most pertinent field evidence. This was then grouped and categorized into core outcomes: (1) neurological development and brain function, (2) NTD risk reductions, (3) reduced risk of adverse pregnancy outcomes, and (4) liver function.

Key search terms used included: “choline” and “pregnancy” OR “lactation”. Publications were restricted to human meta-analysis or systematic review publications. Reference lists of publications were also evaluated to further identify relevant publications. Papers were excluded if: they were not published in the English language, used animal data, focused on choline metabolism rather than choline specifically, or did not report studies in relation to fetal/maternal health outcomes. Publications were restricted to the last 10 years.

## 4. Choline: Latest Evidence

Choline has many roles in fetal development and fetal and offspring health (Figure 3). An evaluation of the latest scientific evidence shows that these include:

Neurological Development and Brain Function—Choline is essential for fetal brain development and shortfalls may hamper this and manifest as behavioral and/or cognitive problems in childhood [34]. A recent meta-analysis and systematic review of 27 publications (19 exploring prevalence and 8 associations) found that inadequate choline intake during pregnancy was associated with an increased risk of developmental impairments in offspring, such as NTDs, cognitive deficits, and altered brain development [35]. Another meta-analysis including five of seven case–control studies concluded that higher maternal choline intakes were likely to be associated with better child neurocognition [36]. Gould et al. (2025) reviewed nine cohort studies and two case–control studies finding that mothers receiving choline supplementation had children with better cognitive development measures [37].

Several systematic reviews have also evaluated field evidence [16,38]. A systematic review of 16 human and 38 animal studies showed that during the first 1000 days of life choline supplementation could reinforce normal brain development resulting in lifelong memory enhancement and protect against metabolic and neural insults [16]. Focusing on neural insults, Akison et al. (2018) concluded that choline supplementation could ameliorate specific behavioral, cognitive, and neurological offspring deficits caused by fetal alcohol exposure, largely using evidence from 18 preclinical studies [38]. Leermakers et al. (2015) using earlier observational evidence also reported that maternal choline intakes could benefit neurological health of the child [39].

NTD Risk Reduction—As choline and folate appear to be metabolically inter-related, maternal imbalances could impact of offsprings’ long-term health [18]. A meta-analysis of five case–control studies [40,41,42,43,44] found that lower maternal choline intakes were associated with a higher risk (36% higher odds ratio; OR) of NTDs which was potentially 2.36-fold higher in some populations [36]. As a wider part of the analysis, it was concluded that higher maternal choline intakes during the second half of pregnancy and early postnatal period demonstrated favorable effects on several domains of child neurocognition, such as memory, attention, and visuospatial learning [36].

Reduced Risk of Adverse Pregnancy Outcomes—A meta-analysis and systematic review of six studies found that higher maternal choline levels were associated with a reduced risk of developing adverse pregnancy outcomes (APOs) (OR 0.51, 95% CI 0.40–0.65) [35]. APOs in this publication included gestational diabetes mellitus, preeclampsia, preterm delivery, and small for gestational age infants [35].

Liver Function—Choline is a methyl donor that enables PtdCho synthesis in the liver and shortfalls can subsequently affect liver lipid transport and cell membrane function [11]. PtdCho is needed for the formation of very low-density lipoprotein that facilitates lipoprotein-mediated transport of triglycerides from the liver into the circulation [33].

The number of pregnant women with fatty liver is projected to rise in the future [45]. Several studies have demonstrated that insufficient choline intake can lead to hepatic fat accumulation, which is reversible upon reintroduction of choline into the diet [11]. This effect is particularly pronounced during pregnancy and lactation as these are periods when choline requirements are elevated [11]. Alongside this, maternal obesity has been shown to be an independent risk factor for NAFLD in offspring [46]. Insufficient maternal choline intake can lead to an accumulation of triglycerides in the liver and subsequent liver dysfunction and risk of fatty liver in both the mother and offspring [11,33]. Research shows that the maternal-to-child choline gradient can deplete maternal choline reserves and increase maternal non-alcoholic fatty liver disease (NAFLD) risk [11]. Fatty liver severity in the offspring depends on maternal fatty liver severity and the duration of exposure to a choline-deficient diet [11].

Overall, a growing body of literature has looked at the roles of choline in fetal development and health. Meta-analysis studies support significant relationships between choline and neurological development [35,36], NTD risk reduction [36], and reduced risk of developing APOs [35]. The exact mechanisms in terms of how choline may exert these actions are not yet fully understood. Equally, ongoing dietary intake studies in different regions and as diets continue to evolve are needed to continuously monitor habitual choline intakes.

**Figure 3 nutrients-17-01558-f003:**
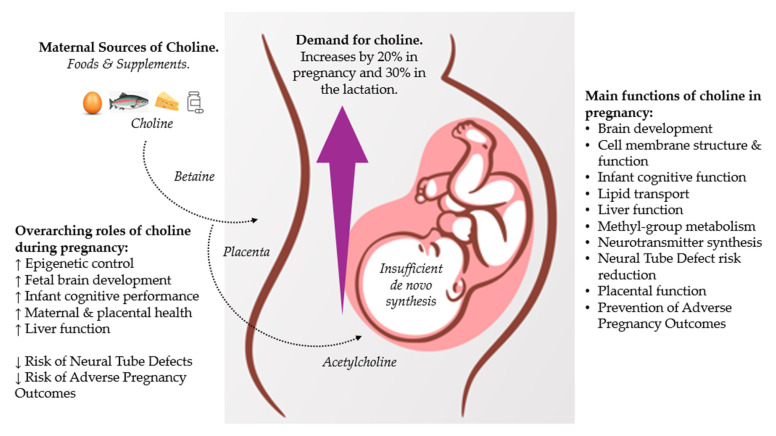
Sources, roles, and functions of choline during pregnancy. Adapted from Caudill et al. (2020) [33,47].

The importance of choline during pregnancy and lactation has been reaffirmed by the American Academy of Pediatrics [48]. Choline is needed for the structural integrity and signaling functions of cell membranes; it is the major source of methyl groups and it directly affects cholinergic neurotransmission, trans-membrane signaling, and lipid transport/metabolism [33,49]. Adequate plasma choline levels are also needed for healthy blood–brain barrier transport of choline [49]. The availability of choline influences neural tube closure and hippocampal development, apoptotic signaling in neurons and liver cells, hepatic transport of lipo-proteins, and hepatic carcinogenesis [24].

In general, inadequate intakes of choline are associated with APOs which include NTDs [36], cleft palates [41], cognitive deficits in offspring [16], and maternal health complications such as gestational diabetes mellitus, preeclampsia, preterm delivery, and delivering small for gestational age infants [35]. Maternal choline intake during pregnancy and lactation can have lasting neurocognitive effects [48].

## 5. Choline: Intake Guidelines and Health Claims

At present there are only formal choline dietary intake recommendations set in the United States, Canada, and Australia for pregnancy and lactation (Table 1) [6,50,51]. Nutrition guidelines from the United States Department of Agriculture [52], United States Food and Drug Administration [53], International Federation of Gynecology and Obstetrics [54], Society of Obstetricians and Gynaecologists of Canada [55], and the Australian Government Department of Health [56] now emphasize the importance of an adequate choline intake during pregnancy and recognize its essential roles in fetal development [35]. The American Academy of Pediatrics [57] also advises pediatricians to move beyond simply advising a “good diet” and that key nutrients including choline can support neurodevelopment and deficits can result in lifelong deficits in brain function. The American Medical Association [58] further advises that prenatal vitamin supplements should contain “evidenced-based” amounts of choline.

In Europe, the growing evidence base that choline is an essential nutrient crucial for fetal health and brain development [16,26,36] is still awaiting translation into population-wide recommendations and policy. Whilst EFSA [33] have established adequate intake (AI) recommendations for choline these have not yet been formally translated into policy. Only the Nordic Council of Ministers [59], which represents Scandinavian countries, has included choline for the first time within their 2023 Nordic Nutrition Recommendations. This reinforces the need for increasing education and awareness among healthcare professionals and prompt translation of available evidence into guidelines, policies, and practice to improve the health of mother and child.

Several EFSA Health Claims [60] for choline have also been authorized which include: (1) “Choline contributes to normal lipid metabolism”, (2) “Choline contributes to normal homocysteine metabolism”, and (3) “Choline contributes to the maintenance of normal liver function”. EFSA also concluded in 2023 that “a cause-and-effect relationship has been established between the intake of choline in pregnant and lactating women and contribution to normal liver function of the fetus and exclusively breastfed infants” [61].

## 6. Choline: Maternal Dietary Intakes

An analysis of global data from 23 studies with women of childbearing age found choline intakes to range from 233 mg to 383 mg/d [62]. In most studies undertaken after 2015 choline intakes did not exceed 80% of the AI. Mean choline intake for adults in different European countries is around 310 mg/day and in non-European countries around 293 mg/day [63]. Over the last decade choline intake amongst adults has been insufficient both inside and outside Europe across populations, including pregnant women [63].

Younger Spanish women of childbearing age (18–30 years) have been found to have lower choline intakes (292 mg/d) compared with older women of childbearing age (31–45 years; 312 mg/d) [64]. Amongst a sample of pregnant German women (n = 283) only 7% achieved the AI for choline with mean intakes being 260 mg ± 141 mg/d [65]. Pregnant vegetarians had significantly lower daily choline intakes that were 205 mg/d compared with omnivores who had daily intakes of 270 mg/d (*p* < 0.0001) [65]. Only 5% of the German pregnant women took dietary supplements containing choline that provided 19% of total choline intake [65].

In Canada the difference between normal-weight and overweight/obese women was statistically significant when choline intake was considered as mg/kg of body mass/day. Furthermore, choline intake was 36% lower in obese women compared to normal weight women [63,66]. Gao et al. (2016) also found that higher choline intakes were associated with a more favorable body composition [66].

These data reflect a concerning situation in that almost all pregnant women have choline intakes that are below the recommended guideline. Choline appears to be an underconsumed and overlooked nutrient, especially in regions where choline intake guidelines have not been established, such as Europe and Asia [35,67]. Further data is presented in Appendix A (Figure A1).

## 7. Choline: Food and Supplement Sources

Choline can be present in foods as both lipid-soluble (PtdCho and sphingomyelin) and water-soluble forms, with most food databases and dietary surveys tending to report data on “total choline” [68,69]. It should be considered that choline availability to cells depends on food intake, absorption, and the efficacy of cellular transport systems [30]. As shown in Figure 4, eggs, meat, and fish have a particularly high choline content although dairy products, legumes, nuts, cruciferous vegetables, and whole grains also provide dietary choline but to a much lower extent [70]. Future choline intakes may depend on trends in egg, meat, and fish consumption which are generally higher in choline compared with plant-based foods which tend to have a lower choline profile [63]. An analysis of an ultra-processed dietary profile found that, whilst some macro- and micronutrients were provided, adequate amounts of choline were not obtained [71].

When choline intakes from the diet are inadequate and fall below recommended intake thresholds, choline supplements may play a role, helping to narrow gaps between habitual intakes and recommendations [35,62,72]. These can include forms such as choline bitartrate, lecithin, citicoline, and choline alfoscerate (C8H20NO6P), also known as alpha-glycerophosphocholine (a-GPC, or GPC) [72]. Choline supplementation during pregnancy and lactation has been regarded as an effective means of closing the nutritional gap between dietary intakes and increased needs during these periods of critical brain development, helping to reinforce cognition in the next generation [26,62,73,74]. Prenatal vitamins can vary widely in their content but are particularly important in instances when dietary intakes fall below recommended intakes, as in the case of choline [75]. Choline supplements are generally considered to be safe. The tolerable upper intake level for choline is 3.5 g/d, which should not be exceeded on a daily basis [6,33].

## 8. Choline: Application for Clinical Practice

Clinicians are well placed to support mums-to-be with evidence-based nutritional care. Scientific evidence for maternal nutrition has now evolved far beyond simply recommending folic acid, as agreed by international expert consensus [2]. The value of nutrition across the first 1000 days of life (conception to 2 years into infancy) is well recognized for its importance in relation to neurodevelopment and lifelong mental wellbeing [9,16,57,76]. Research on choline roles and needs during pregnancy and lactation is now well established. As choline is crucial for brain development, neural tube formation, DNA and histone methylation, liver function, and the overarching wellbeing of expectant mothers, it is a fundamental cornerstone of prenatal care [11,70,77].

Given that many women of childbearing age are underconsuming choline it is vital that all women, regardless of social status, are aware of its importance [70]. Clinicians are well placed to disseminate this advice in an evidence-based fashion. Whilst the diet can provide some choline certain factors can contribute to choline shortfalls (Table 2). In these instances, choline supplementation in dosages appropriate to close nutritional gaps should be considered. Regarding prenatal vitamins, the nutritional profile of these can vary widely but when recommended these should include choline as intakes are generally below recommended intakes during pregnancy and lactation [75]. 

## 9. Conclusions

Despite the fact that science has accrued since 1998, choline intake inadequacies amongst women of childbearing age remain a public health concern, especially during pregnancy and lactation. Choline deficiencies can have severe ramifications for the developing fetus and breastfed infant, including neurodevelopmental outcomes, NTDs, and NAFLD [11,16,81]. Certain foods such as eggs, oily fish, meat, and poultry can provide good dietary sources of choline [70]. However, it should be considered that the physiological demands for choline during pregnancy and lactation are some of the highest across the lifespan and subsequently gaps between habitual dietary choline intakes and recommendations substantially widen. Considering this, guidelines on maternal micronutrient recommendations need to be revised and supplementation strategies that include choline should be considered by clinicians.

## Figures and Tables

**Figure 1 nutrients-17-01558-f001:**
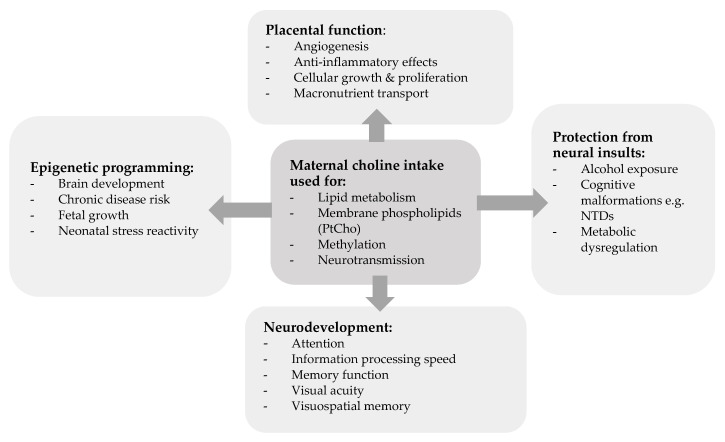
An overview of maternal choline intake for metabolic processes. Adapted from Korsmo et al. (2019) [17].

**Figure 2 nutrients-17-01558-f002:**
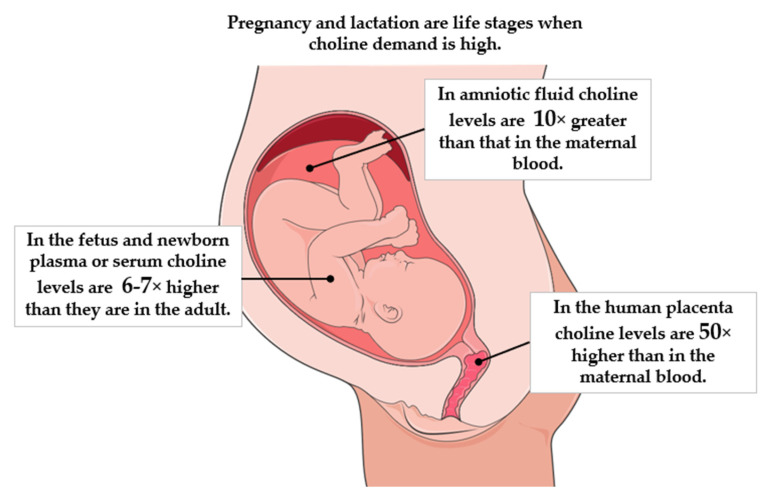
The fetus is living in a high-choline environment. Data extrapolated from: [26,27,28,29,30,31]. Image from SMART Servier Medical.

**Figure 4 nutrients-17-01558-f004:**
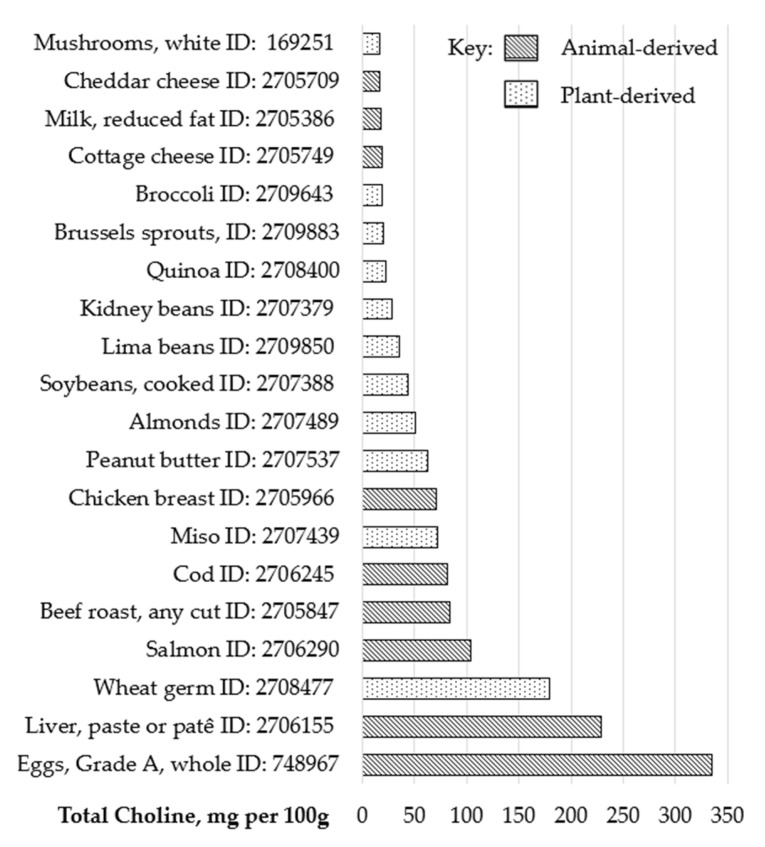
Choline content of animal-derived and plant-derived food sources—Total Choline, mg per 100 g. Source: USDA database [68].

**Table 1 nutrients-17-01558-t001:** Choline Intake Recommendations.

US IOM (1998) [6] AI (mg/d)		Canada (2023) [50] AI (mg/d)	Australia (2006) [51] AI (mg/d)	EFSA (2016) [33] AI (mg/d)	NNR (2023) [59] AI (mg/d)
Life Stage	Females ≥19 yrs	Females ≥19 yrs	Females ≥19 yrs	Females ≥18 yrs	Females ≥18 yrs
Adult	425	425	425	400	400
Pregnancy	450	450	440	480	-
Lactation	500	550	550	520	-

Key: AI, Adequate Intake; EFSA, European Food Safety Authority; IOM, Institute of Medicine; NNR, Nordic Nutrition Recommendations.

**Table 2 nutrients-17-01558-t002:** Evidence-based guidance for Clinical Practice.

	Evidence-Based Guidance Points
** *Importance of choline for pregnancy and lactation* **	– Brain development – Cell membrane structure and function (phosphatidylcholine) – Fetus, preterm infants and full-term newborns rely on maternal choline supply – Infant cognitive function – Lipid transport – Liver function – Methyl-group metabolism – Neurotransmitter synthesis (acetylcholine) – Neural tube defect risk reduction – Normal homocysteine and lipid metabolism – Placental function – Prevention of adverse pregnancy outcomes
** *Choline food sources* **	– *Animal-derived* choline-rich food sources such as eggs, meat and poultry, fish, and dairy products [68] – *Plant-derived* legumes and nuts (e.g., soybeans and peanuts), cruciferous vegetables (e.g., broccoli and Brussels sprouts), and whole grains, such as quinoa, although these tend to be lower in choline [68]
** *Choline requirements* **	– Pregnancy and lactation are periods when maternal reserves of choline are depleted due to increased needs to support fetal/infant growth and development [24,26] – Choline needs increase by 20% during pregnancy and 30% during lactation [33] – EFSA [33] recommends: 480 mg/d choline for pregnancy and520 mg/d choline for lactation
** *Factors contributing to higher choline requirements* **	– Pregnancy and lactation (high demands) – Newborns and infants (low endogenous choline synthesis at birth but high demands) – Low dietary intake of folate, vitamin B12 (impairs de novo choline synthesis) [78] – Polymorphisms in phosphatidylethanolamine *N*-methyltransferase (PEMT) gene (low de novo choline synthesis) [79] – High-fat diet and high sugar intake (heightened risk of non-alcoholic fatty liver disease) [80] – Plant-based diets (lower choline intake from foods) [65] *These factors increase susceptibility to insufficient choline status*
** *Choline dietary intakes* **	– EFSA reports habitual choline intakes of 356 mg/d in pregnant women—this is even lower in some countries [18,51] – Most women do not meet choline intake recommendations from the diet
** *Consequences of inadequate choline intakes during pregnancy and lactation* **	↑ Risk of impairments in brain development [16,26] ↑ Risk of cognitive deficits [16,36] ↑ Risk of neural tube defects [36] ↑ Risk of adverse pregnancy outcomes such as gestational diabetes mellitus, preeclampsia, preterm delivery, and small for gestational age infants [35]
** *Supplementation guidance* **	– Choline supplement use is recommended to close nutritional gaps when dietary choline intakes are not sufficient

Source: Adapted from Jaiswal et al. (2023) [70].

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
