# Peer review of "Choline in Pregnancy and Lactation: Essential Knowledge for Clinical Practice"

_nutrients, 2025, doi:10.3390/nu17091558_

Round 1
Reviewer 1 Report
Comments and Suggestions for Authors
The manuscript is very interesting, and the author provides an updated review on the nutritional importance of choline during pregnancy and breastfeeding. The manuscript is well-written, and the information the author presents may be useful for research, especially for clinical professionals. However, I have the following comments:
I. Comments:
1. Improve the wording of the review objective.
2. After the introduction, develop a section on the criteria used to select the cited papers.
3. Regarding choline, improve the following aspects:
3.1. Choline metabolism
3.2. Describe the mechanisms in which choline may be involved, especially in growth and development.
4. Have alterations in choline metabolism been observed in maternal obesity?
5. Currently, the Western diet is characterized by high energy density and unhealthy foods. In this context, has an association been observed between obesity, unhealthy eating, and choline deficiency? 6. Improve aspects of the use of choline supplements during pregnancy. The author could refer to studies that have used choline and other nutrients as supplements.
7. Improve the resolution of figures.
Author Response
Reviewer 1
The manuscript is very interesting, and the author provides an updated review on the nutritional importance of choline during pregnancy and breastfeeding. The manuscript is well-written, and the information the author presents may be useful for research, especially for clinical professionals. However, I have the following comments:
Comments:
Improve the wording of the review objective. Done – reworded to The objective of the present article is to provide a coherent update of the latest
After the introduction, develop a section on the criteria used to select the cited papers. A new approached section has been added - A PubMed and Science Direct search for human clinical evidence restricted to the last 10 years (meta-analysis and systematic/review publications) was undertaken to identify the latest, most pertinent field evidence. This was then grouped and categorized into core outcomes: 1) neurological development and brain function, 2) NTD risk re-ductions, 3) Reduced risk of adverse pregnancy outcomes and 4) Liver function.
Regarding choline, improve the following aspects:
3.1. Choline metabolism A metabolism section has been added and a figure
3.2. Describe the mechanisms in which choline may be involved, especially in growth and development. A metabolism section has been added and a figure
Have alterations in choline metabolism been observed in maternal obesity? This is described under section 4 and section 6 now.
Currently, the Western diet is characterized by high energy density and unhealthy foods. In this context, has an association been observed between obesity, unhealthy eating, and choline deficiency? The maternal dietary intake section explains how choline is under consumed in western regions. More has been added into this section. In Canada the difference between normal-weight and overweight/obese women was statistically significant when choline intake was considered as mg/kg of body mass/day. Furthermore, the decline in choline intake was 36% in obese women compared to normal weight women[63; 66]. Gao et al. (2016) also found that higher choline intakes were associated with a more favourable body composition[66].
In section 7 is has been added that UPFs have been associated with lower choline intakes/shortfalls.
Improve aspects of the use of choline supplements during pregnancy. The author could refer to studies that have used choline and other nutrients as supplements. This has been included under section 7
Improve the resolution of figures. These have been improved.
Reviewer 2 Report
Comments and Suggestions for Authors
This is an interesting and well-organized review article with adequate novelty. Some points should be addressed.
- Subheadings (Bacground/Objectives, Methods, Results, Conclusions) should be added in the Abstract.
- More information should be added in the Introduction section.
- At the end of the Introduction section and before the ain of the study, the authors should emphasize the literature gap that their article aims to cover.
- After the Introduction section, the authors should add a Method section reporting the databases used and the relevant keywords used to collect their data. Exclusion and inclusion criteria should also be mentioned.
- In line 84 "A recent meta-analysis and systematic review of 27 publications found ......". Please specify what type of studies were included in this systematic review.
- In line 89 ''Gould et al. (2025) identified clinical trials....). Please specify what type of clinical studies were included in ths systematic review.
- In line 103-107, please specify what type of clinical studies were includes in this systematic review-meta-analysis.
- In patagraph in lines 108-117, are there any study for overweight or obese prergnant women with increased fat content in the liver?
- The authors should add a Figure describing the potential mechanisms through which choline affect fetal development.
- The resolution of Figure 3 should be improved.
- At the end of the Discussion section, the authors should discuss the strenghts and the limitation of the existing studies concerning choline biological properties in pregnancy.
Author Response
Reviewer 2
This is an interesting and well-organized review article with adequate novelty. Some points should be addressed.
Subheadings (Background/Objectives, Methods, Results, Conclusions) should be added in the Abstract. These have been added
More information should be added in the Introduction section. More on choline metabolism has now been added into the introduction.
At the end of the Introduction section and before the aim of the study, the authors should emphasize the literature gap that their article aims to cover. Within the scientific literature few articles are written with clinicians in mind which was the aim of the present article.
After the Introduction section, the authors should add a Method section reporting the databases used and the relevant keywords used to collect their data. Exclusion and inclusion criteria should also be mentioned. This has been addressed
In line 84 "A recent meta-analysis and systematic review of 27 publications found ......". Please specify what type of studies were included in this systematic review. 19 exploring prevalence and eight associations
In line 89 ''Gould et al. (2025) identified clinical trials....). Please specify what type of clinical studies were included in ths systematic review. Gould et al. (2025) reviewed nine cohort studies and two case-control studies finding in instances that mothers
In line 103-107, please specify what type of clinical studies were includes in this systematic review-meta-analysis. They were case-controlled studies. This is included.
In patagraph in lines 108-117, are there any study for overweight or obese prergnant women with increased fat content in the liver? It has been added that the number of pregnant women with fatty liver is expected to increase in the future.
The authors should add a Figure describing the potential mechanisms through which choline affect fetal development. This has been added – Figure 1
The resolution of Figure 3 should be improved. This has been improved
At the end of the Discussion section, the authors should discuss the strenghts and the limitation of the existing studies concerning choline biological properties in pregnancy. An additional section has been added at the end of section 4….Overall, a growing body of evidence has looked at the roles of choline in fetal development and health. Meta-analysis studies support significant relationships between choline and neurological development [35; 36], NTD risk reduction[36] and reduced risk of developing APOs[35]. The exact mechanisms in terms of how choline may exert these actions are not yet fully understood and will need to continue. Equally, ongoing dietary intake studies in different regions and as diets continue to evolve are needed to continuously monitor habitual choline intakes.
Reviewer 3 Report
Comments and Suggestions for Authors
Manuscript ID: nutrients-3602588
Type of manuscript: Article
Title: Choline in Pregnancy and Lactation: Essential Knowledge for Clinical
Practice.
The present article aims to provide a coherent update of the latest clinical evidence, choline guidelines, health claims and practical advice for clinicians.
Comments and Suggestions for Authors:
The manuscript is interesting but requires some considerations.
The manuscript is not an article but a narrative review. This should be stated in the Title and Abstract sections.
The manuscript indicates that Science has been written succinctly, but what does this succinct review contribute compared to other broader and more systematic reviews with meta-analysis, such as the recent meta-analysis and systematic review of 27 publications by Nguyen et al. Choline in pregnant women: a systematic review and meta-analysis. Nutr Rev (2025)?
Figures are displayed in poor quality.
Although this is a narrative review, the inclusion and exclusion criteria used in its implementation should be included.
Table 1 shows the American and European choline intake recommendations but includes those from Australia. This should be corrected.
References should be thoroughly revised to conform to uniform and appropriate standards for the journal Nutrients.
Author Response
Reviewer 3
Manuscript ID: nutrients-3602588
The present article aims to provide a coherent update of the latest clinical evidence, choline guidelines, health claims and practical advice for clinicians.
Comments and Suggestions for Authors:
The manuscript is interesting but requires some considerations.
The manuscript is not an article but a narrative review. This should be stated in the Title and Abstract sections. This has been stated/included now above the title – narrative review article and in the abstract but would make the article title rather long. It is now clear that is it a narrative review.
The manuscript indicates that Science has been written succinctly, but what does this succinct review contribute compared to other broader and more systematic reviews with meta-analysis, such as the recent meta-analysis and systematic review of 27 publications by Nguyen et al. Choline in pregnant women: a systematic review and meta-analysis. Nutr Rev (2025)? This review provides core evidence that is easier for clinicians to extract – it has infographics, a choline intake recommendations table and evidence-based guidance for clinical practice specifically.
Figures are displayed in poor quality. These have been enlarged/improved
Although this is a narrative review, the inclusion and exclusion criteria used in its implementation should be included. This has been added.
Table 1 shows the American and European choline intake recommendations but includes those from Australia. This should be corrected. The title has been updated.
References should be thoroughly revised to conform to uniform and appropriate standards for the journal Nutrients. Endnote software has been used and input using the Frontiers option. References have been reprogrammed if a citation was omitted by the software.
Round 2
Reviewer 1 Report
Comments and Suggestions for Authors
Author answered all my questions and comments. The manuscript was improved. Therefore, the manuscript can be accepted.
Reviewer 2 Report
Comments and Suggestions for Authors
The authors have significantly improved their manuscript.